# Hyaluronic Acid Interacting Molecules Mediated Crosstalk between Cancer Cells and Microenvironment from Primary Tumour to Distant Metastasis

**DOI:** 10.3390/cancers16101907

**Published:** 2024-05-16

**Authors:** Yali Xu, Johannes Benedikt, Lin Ye

**Affiliations:** 1Cardiff China Medical Research Collaborative, Division of Cancer and Genetics, Cardiff University School of Medicine, Cardiff CF14 4XN, UK; xuy109@cardiff.ac.uk; 2School of Engineering, Cardiff University, Cardiff CF24 3AA, UK; benedikt@cardiff.ac.uk

**Keywords:** hyaluronic acid, hyaluronic acid interacting molecules, tumour microenvironment, cancer, metastasis, adhesion, invasion and epithelial mesenchymal transition

## Abstract

**Simple Summary:**

Hyaluronic acid (HA) is a kind of polysaccharide that widely exists in tissues of vertebrates, functioning as a lubricator, an adhesion-reducing molecule, and an extracellular matrix (ECM) organiser. The function of HA depends on its relative molecular mass. High molecular weight HA (HMW-HA) helps to maintain tissue homeostasis, and can be degraded to elicit pro-inflammatory, angiogenic, and tumour microenvironment (TME) remodelling effects. HA accumulation was found to be linked to cancer progression and worse patient outcomes. The term hyaluronic acid-interacting molecule (HAIM) refers to a group of proteins that can interact with HA and participate in the regulation of cancer cells and immune cell behaviour. Membrane-anchored HAIMs bind to HA as receptors to regulate cellular function through various pathways, while secreted HAIMs bind with HA as a scaffold to construct ECM and can also act on a variety of cells in the TME to coordinate release and actions of cytokines and enzymes to reshape the ECM. Some of these molecules also play a role in HA transport, endocytosis, and degradation. Understanding these complex and profound interactions between HAIMs and HA will shed light on their role in the TME to foster a supportive microenvironment for local invasive expansion and distant spread of cancer cells.

**Abstract:**

Hyaluronic acid (HA) is a prominent component of the extracellular matrix, and its interactions with HA-interacting molecules (HAIMs) play a critical role in cancer development and disease progression. This review explores the multifaceted role of HAIMs in the context of cancer, focusing on their influence on disease progression by dissecting relevant cellular and molecular mechanisms in tumour cells and the tumour microenvironment. Cancer progression can be profoundly affected by the interactions between HA and HAIMs. They modulate critical processes such as cell adhesion, migration, invasion, and proliferation. The TME serves as a dynamic platform in which HAIMs contribute to the formation of a unique niche. The resulting changes in HA composition profoundly influence the biophysical properties of the TME. These modifications in the TME, in conjunction with HAIMs, impact angiogenesis, immune cell recruitment, and immune evasion. Therefore, understanding the intricate interplay between HAIMs and HA within the cancer context is essential for developing novel therapeutic strategies. Targeting these interactions offers promising avenues for cancer treatment, as they hold the potential to disrupt critical aspects of disease progression and the TME. Further research in this field is imperative for advancing our knowledge and the treatment of cancer.

## 1. Hyaluronic Acid 

Hyaluronic acid (HA), also known as glycosaminoglycan (GAG), is a linear unsulphated polysaccharide widely found in the vertebrate extracellular matrix (ECM). Alternating β-1,4 and β-1,3 glycosidic bonds link D-glucuronic acid (GlcUA) acid and N-acetylglucosamine (GlcNAc) into repeating units. HA ranges in size from as short as a single disaccharide unit to as long as 25,000 disaccharide units, which results in the polydispersity of HA in vivo with various molecular weights ranging from a few kilodaltons (kDa) to several thousand kDa. Unlike other glycosaminoglycans, instead of being synthesised intracellularly in Golgi and subsequently secreted, HA is produced by HA synthase isoenzymes (HAS1–3) located on the plasma membrane, with simultaneous extrusion to the extracellular space [1]. 

Hyaluronic acid can be found in the tissues and the bodily fluids of vertebrates, and even in certain bacterial species [2]. HA is notably abundant in various tissues, constituting a significant portion of tissues like skin and cartilage, with the most abundance in the foetal umbilical cord and joint synovial fluid within the human body [2]. The dermis and vitreous body contain hundreds of micrograms of HA per gram [2]. HA is found on the surfaces of bodily tissues that undergo sliding movements. Its exceptional lubricating qualities have been demonstrated to effectively reduce the formation of adhesions following surgical procedures in the abdomen and musculoskeletal system. It is also commonly found within the ECM in all vertebrate organisms. Specialised cells like fibroblasts, keratinocytes, and chondrocytes play crucial roles in the continuous production and release of HA throughout the body. HA further exhibits scavenging abilities against free radicals and cellular debris [3,4]. Additionally, other unique characteristics of HA, including its biocompatibility, degradability, and non-immune activating properties, make it suitable as a drug carrier [5,6]. 

HA levels were found to be significantly increased in breast cancer, prostate cancer, ovarian cancer, endometrial cancer, gastric cancer, lung cancer, and pleural mesothelioma, exhibiting a significant correlation with malignancy and serving as an independent unfavourable prognostic marker (Table 1). The elevated HA levels in these tumours may result from the hyperactivation of HA synthase isoenzymes (HAS) and hyaluronidase (HYAL) expression, or from a reduced turnover of HAS protein rather than an increase in the expression of HAS at the gene level [7]. However, it remains unclear whether these enzymes are responsible for creating and maintaining particular HA polymer sizes regarding synthesis or degradation.

HA accumulation is detected in the tumour stroma, pericellular area, and blood serum [8,9,10,11]. Since certain HAIMs are overexpressed by tumour cells or enriched in the tumour microenvironment (TME), HA has been utilised for delivering drugs to tumours, and this was reviewed recently [12].

The concentration and average molecule weight of HA are regulated by both its synthesis and degradation processes, and its biological function is heavily dependent on its dimension. Nevertheless, high-molecular-weight HA (HMW-HA) and low-molecular-weight HA (LMW-HA), including HA fragments, have distinct roles in the human body. The role of different sizes of HA has been reviewed by Robert and his colleague [13]. HMW-HA (>250 kDa) is a space-filling molecule with hydration capacity that allows us to retain water and create a supporting framework for cells. HMW-HA also functions as a lubricant and a liquid shock absorber in the articular cavity. Furthermore, it can also regulate intracellular signal transduction upon binding to cell membrane receptors like CD44 [14], thereby affecting various cell biological processes, including anti-angiogenesis, inhibition of cell proliferation, suppression of immune responses, resistance to inflammatory responses, promotion of tissue integrity and quiescence, inhibition of phagocytosis, and the synthesis of HA itself [15,16]. In contrast, the accumulation of LMW-HA (<250 kDa) and HA oligosaccharides (<10 kDa) has been linked to the aggressiveness traits in cancer cells, as it promotes angiogenesis, inflammatory responses, and immune responses. It further induces the production of specific cytokines and enzymes necessary for the constitution of the tumour microenvironment (TME). Specifically, HA molecules around 200 kDa are known to trigger the release of inflammatory chemokines and impede the process of fibrinolysis [17,18]. Conversely, HA fragments smaller than 10 kDa enhance angiogenesis, facilitate cell migration, support the differentiation of endothelial cells, and also stimulate cytokine production in dendritic cells [13]. Additionally, HA fragments consisting of 4–6 oligosaccharides (0.8–1.2 kDa) have been shown to activate the transcription of matrix metalloproteinases (MMPs) in both tumour cells and primary fibroblasts, thereby aiding in tumour progression [19]. In response to stress, tetrasaccharide chains of HA are capable of upregulating heat shock protein 72, thus preventing cell death [20]. In contrast to the promoting effect of HMW-HA on CD44 clustering, HA oligosaccharides disrupt this procedure [14]. These HA fragments often act as alarms that transmit “danger signals” in vivo [16]. Within the TME, HA with varying molecular weights binds to distinct HAIMs and regulates activities of cancer cells, immune cells, and tumour-associated fibroblasts to create a favourable TME. 

**Table 1 cancers-16-01907-t001:** Aberrant HA levels in cancers.

Cancer Types	HA Category	Clinical Relevance	References
Breast cancer	Serum LMW HA	Lymph node metastasis and angiogenesis	[9]
	Plasma HA	Tumour progression, poor prognosis, worse response to treatment	[21]
	Stromal HA and malignant cell-associated HA	HER2 positivity; elevated tumour size, tumour grading lymph nodes involvement, body mass index and relapse rate; reduced hormone receptor expression, tumour differentiation, overall survival	[22]
			[23]
Colorectal cancer	HA fragments	Early development, cancer progression, lymph nodes metastasis.	[11,24]
	Pericellular HA	Enhanced invasive capacity.	[25]
	HA level	Predictors for OS and DFS	[26]
Ovarian cancer	Pericellular HA	Malignancy; an independent adverse predictor for OS.	[10]
	HA accumulation	Poor differentiation, metastasis, and aggressive phenotype	[27]
	Serum HA before chemotherapy	Chemotherapy resistant; shortened OS and DFS	[28,29]
Endometrial cancer	HA level	Enhanced invasion, tumour grading and lymphatic involvement	[30]
	Tumour stromal HA	Tumour development	[31]
	Peritumour stroma HA	Tumour grade and invasion	[32]
Gastric cancer	HA level	Lymph node metastasis, cancer subtype specific, worse survival outcome.	[33,34]
	Tumoral HA	Cancer-subtype specific	[35]
	Serum HA	Elevated in gastric cancer	[36]
Lung cancer	Baseline plasma HA	Bone metastasis; chemotherapy efficacy	[37]
	HA content	Histological subtype specific, tumour differentiation, stage, recurrence and DFS	[38]
	Tumour HA	Level of malignancy, angiogenesis, patient survival, reflected in sputum	[8]
Mesothelioma	HA level	Increased in pleural fluid	[39]
	High effusion HA level	Better survival	

## 2. HA Interacting Molecules

The extensive range of functional capabilities arises from the considerable variety of proteins that bind to HA. These proteins can be present within cells, can be released, or can be situated on the cell membrane (Figure 1). These molecules include HA receptors CD44, lymphatic vessel endothelial hyaluronan receptor 1 (LYVE1), the receptor for HA-mediated motility (RHAMM), and other HA-interacting molecules, namely TNF-stimulated gene-6 protein (TSG-6), inter-alpha-trypsin inhibitor (IαI) family, the hyaluronan and proteoglycan binding link protein (HAPLN) family, plasma hyaluronan-binding protein (PHBP) and the lecticans, chondroitin sulphate proteoglycan protein (CSPG) families [40,41,42]. Extracellular hyaluronan-binding proteins such as CD44, LYVE-1, TSG-6, and the lectican family members belong to the link protein superfamily, as they all contain a conserved linker module of approximately 100 amino acids (aa) in length which allows them to bind HA. The link proteins, consisting of one immunoglobulin domain and two contiguous link modules, are vital components of the cartilage matrix. The proteins facilitate tissue load-bearing resist to deformation by linking HA and proteoglycans to form large, hydrated, multi-molecule aggregates [43].

In general, all the HAIMs are synthesised in the endoplasmic reticulum (ER) and processed in the Golgi apparatus. CD44 is primarily localised on the cell surface as a transmembrane glycoprotein and is often found in lipid rafts and cell–cell adhesion sites [44], whereas the cell surface receptor LYVE-1 is primarily found on lymphatic endothelial cells, which are primarily localised to the cell membrane [45]. RHAMM can be found both intracellularly and on the cell surface, depending on the cell type and context [46]. The IαI family is composed of a light chain which is associated with one of five homologous heavy chains. Removal of segments from the C-terminal generates matured IαI proteins. Most of the IαI family members are secreted proteins, but some have also been found to be located intracellularly [47]. Other molecules like TSG-6 and the extracellular matrix proteoglycan, including HAPBs, HAPLN family and the lectican family, the latter comprising Neurocan (NCAN), Brevican (BCAN), Aggrecan (ACAN), and Versican (VCAN), are secreted into the ECM [48,49,50,51]. CD44, RHAMM, PHBPs, and HAPLNs are widely expressed in various tissues and cell types, including certain cancer cells. Meanwhile, LYVE-1 is primarily expressed in lymphatic ECs [45], members of the IαI family are synthesised in the liver [52], and ACAN is synthesised primarily by chondrocytes and other cartilage cells [53]. Additionally, NCAN and BCAN are mainly expressed in the central nervous system (CNS) [51] while VCAN is predominantly expressed in loose connective tissue, epithelia, smooth muscle cells, and neural tissues [54]. Notably, TSG-6 is produced by immune cells and stromal cells [55] while ITIH proteins are generated by neoplastic cells and participate in various biological processes, encompassing inflammation, cellular malignancy, and tumourigenesis [56]. The function domains of HAIMs and their interactions with HA are shown in Figure 2 and Figure 3.

CD44 comprises seven extracellular domains, one transmembrane domain, and one cytoplasmic domain. The membrane and the C-terminal cytoplasmic domains remain highly conserved domains. The extracellular BX_7_B motif is responsible for the binding to HA, a function shared by other HA-binding proteins like RHAMM [57]. Additionally, the presence of intramolecular disulphide bonds is crucial for the HA-binding activity [57].

Notably, alternative splicing results in various CD44 isoforms. CD44v variants, in contrast to CD44s, are only present in specific epithelial cells during embryonic development, during the maturation and activation of lymphocytes, and in various carcinoma fields [58]. These variants undergo further modifications through N- and O-linked glycosylation. Nevertheless, the HA-binding domain located in the amino-terminal region is a common feature across all isoforms of CD44. HA produced by stromal cells and cancer cells is a major ligand for CD44. When HA binds to the ligand-binding domain of CD44, it induces a conformational change, allowing adaptor proteins or cytoskeletal elements to bind to the intracellular domain of CD44. This interaction activates various downstream pathways, thereby regulating cell migration, invasion, proliferation, and adhesion [59].

LYVE-1 is a homologue of the CD44 glycoprotein with the ability to bind both soluble and immobilised HA. LYVE-1-mediated HA endocytosis is involved in the degradation of HA by lymphatic endothelial cells. The activation state of the LYVE-1 receptor may depend on the involvement of sialyation receptors and in vitro glycan modifications [60], which is similar to the regulation of CD44-HA binding in blood vessels. However, the exact mode of activation remains unknown. LYVE-1 has been detected in the lymphatic vessels surrounding lymph nodes [61].

More than 80% of HA in tissues is degraded in lymph nodes. LYVE-1 appears to act as a key receptor that is responsible for HA uptake and transportation, mediating its end catabolism within lymphatic ECs or transporting it to the lumen of afferent lymphatic vessels for degradation and reabsorption. However, LYVE-1 mediates endocytosis of HMW-HA in fibroblasts in a way that is different from CD44 [62]. 

Similar to CD44, RHAMM also presents splicing variants and its protein isoforms are phosphorylated by serine–threonine kinases, such as protein kinase C, AURKA, and ERK1/2 [63]. RHAMM expression is cell cycle-related, with both overexpression and lack of RHAMM resulting in genomic instability, thereby promoting tumour progression [64]. The diversity in the subcellular localisation of RHAMM isoforms lead to variations in their specific functions and downstream signalling pathways, which eventually play different roles in cancer [65]. RHAMM does not contain a link module domain. Instead, it possesses a HA-binding region located at its C-terminus, which is facilitated by the BX7B motif. Although it lacks a transmembrane domain, RHAMM is anchored to the cell membrane via glycosyl-phosphatidylinositol (GPIs) and works with CD44 to participate in processes such as cell adhesion, motility, proliferation and differentiation, wound healing, tissue remodelling, modification of Ras signalling and other cellular activities, as well as inflammatory responses [57]. CD44 and CD44-EGFR complexes are RHAMM extracellular binding partners [66,67] while intracellular RHAMM can bind directly to actin filaments and microtubules in the cytoskeleton or indirectly regulate microtubule and centrosome motility [68]. CD44 and RHAMM also exhibit compensatory effects; thus, targeting HA receptors for cancer therapy may be necessary to silence both receptors to completely abolish the corresponding HA signalling. 

PHBP was originally identified as a novel extracellular serine protease in human plasma with similar amino acid sequence to hepatocyte growth factor activator (HGFA) [48], and a similar domain to urinary plasminogen activator (u-PA). In human plasma, PHBP predominantly targets fibrinogen and fibronectin, cleaving both the α and β chains of fibrinogen and HC2 in IαI [48]. PHBP also binds to HA, an interaction which has been being implicated in cancer progression [69]. Most of the HA in the body is HMW-HA, which can inhibit the protease activity of PHBP, while the LMW-HA produced during inflammation can activate PHBP [70].

In mammals, the bloodstream transports many glycoproteins that function as protease inhibitors. These proteins, along with related molecules, constitute the inter-alpha-trypsin inhibitor (IαI) family, which encompasses a range of plasma protease inhibitors. They can generate precursor polypeptides with distinct fates and functions, as well as various intramolecular interchain glycosaminoglycan linkages [47]. In the process of IαI protein maturation, heavy chains ITIH1-3 precursors are assembled with light-chain bikunin and are modified post-translationally. However, the conserved modification site is absent in ITIH4, which is incapable of binding to bikunin [47]. Notably, the heavy chains, primarily ITIH1 and ITIH2, are connected to bikunin by a solitary chondroitin sulphate chain [71,72]. This characteristic renders ITI an exceptional proteoglycan, both in terms of its structure and its function. Its plasma protease inhibitory capabilities [72] are exclusively confined within the bikunin component of the molecule [47]. For the heavy chains, the only known function so far is their covalent binding to HA, which earns them an additional designation of serum-derived hyaluronan-associated protein (SHAP). Their transferring and binding to HA requires the presence of a functional regulator called TNF-stimulated gene (TSG-6), also known as tumour necrosis factor alpha-induced protein 6 (TNFAIP6). During the transesterification reaction, TSG-6 and HA form a stable complex to regulate HA structure and inhibit proteases by acting as a catalytic factor to promote the transfer of heavy chains with an assistance by calcium ions [73]. Moreover, TSG-6 has been found to enhance the anti-plasmin activity of ITI [74].

Upregulation of TSG-6 has been evident in kidney cancer, ovarian cancer, and stomach cancer [41]. Two different homologous regions at the C-terminal and N-terminal ends of TSG-6 form different structural domains, which are present in a variety of proteins with similar modular structures such as CD44, NCAN, ACAN, and VCAN [55,75]. The N-terminal domain of TSG-6 is a structural motif with a HA-binding affinity [76]. HA-TSG-6 interaction induces the formation of TSG-6 oligomers which facilitates crosslink among individual HA molecules and HA condensations. The strong interaction between the TSG-6 HA-binding module and HA has been used to develop a histological method for detecting HA [77]. The C-terminal half of TSG-6 has a structure whose definition is entirely based on sequence homology and the conservation of certain structural elements; this is known as the evolutionally conserved domain for complement C1r/C1s, Uegf, Bmp1 (CUB) [78,79]. It is unique in that TSG-6 simultaneously possesses the link module and CUB domains. The link module of TSG-6 protein is similar to that of CD44. This similarity implies that the positions of the interaction surfaces may also be conserved in other proteins containing the link module, although the specific molecular conformation in binding to HA may differ. Furthermore, an interaction between TSG-6 and CD44, expressed by the M1 macrophage, enables CD44 to form a complex with HA. This complex formation prevents CD44 from interacting with Toll-like receptor (TLR), thereby inhibiting the downstream NF-κB pathway and the secretion of pro-inflammatory cytokines [80]. 

HAPLNs and lecticans also belong to the link protein superfamily. The amino acid sequences encoded by the four HAPLN family members share a similarity of 45–52%. In terms of genome structure, their exon–intron organisation is also similar to the 5’ exon genomic organisation of the core protein of the lectican family [81]. Remarkably, the four HAPLN genes are located close to their corresponding four lectican core protein genes, forming four sets of lectican-HAPLN genes within the mammalian genome [81]. Each of the four HAPLN family members contains an N-terminal signal sequence, an immunoglobulin (Ig) domain, and two link modules. These link proteins bind to lecticans via the Ig domain, and the low sequence identity of this domain indicates a binding specificity of individual HAPLNs to lecticans [81]. 

All the lectican family members have a link protein structure comprising an Ig domain followed by two link modules, also known as G1 domain, at their N terminal, and at least one EGF module followed by a C-type lectin module and an OSF-2 domain at their C-terminal, which is also known as the G3 domain. The length of glycosaminoglycan attachment domain varies among the family members. Although the N-terminal is responsible for HA binding, the structural integrity of the C-terminal domain of the NCAN core protein determines the ability of cells to adhere to the extracellular matrix [82]. When NCAN was recombinantly expressed in HEK-293 cells, the presence of NCAN resulted in a detachment of the producing cells which formed suspending spheroids; both the NCAN-containing chondroitin sulphate chains and the molecule’s C-terminal are required to achieve this. In contrast, cells secreting the N-terminal domain of NCAN, which binds to HA, was demonstrated increased adhesiveness [82]. VCAN regulates cellular function mainly through its G1 and G3 domains. G1 domain promotes migration but inhibits adhesion [83]. The cell migration process is facilitated by both the core protein domain and glycosaminoglycan side chains of VCAN [84]. Specifically, the G3 domain of VCAN can promote endothelial cell proliferation, adhesion, and migration [85]. Moreover, acting as a mitogen, it can not only promote cell proliferation but also inhibits apoptosis by activating EGFR through the EGF-like motif within the G3 domain [86]. 

## 3. Deregulated HAIMs in Solid Tumours

Abnormal expression of HAIMs has been detected in various solid tumours, which is related to tumour progression, prognosis, and chemoresistance (Table 2) [41,56,87,88,89,90,91,92,93,94,95,96,97,98,99,100,101,102,103,104,105,106,107,108,109,110,111,112,113,114,115,116,117,118,119,120,121,122,123,124,125,126,127,128,129,130,131,132,133]. ITIHs, LYVE-1, RHAMM, HAPLN1, and PHBP levels are elevated in certain tumour types, while decreased expression of these HAIMs has been evident in other malignancies [41,56,92,105,110,111,115,116,125,126]. For instance, elevated expression of ITIH2 in liver hepatocellular carcinoma (LIHC) patients was associated with longer overall survival (OS) (*p* = 0.019) [90], while increased expression of ITIH2 was associated with shorter OS in colorectal cancer patients that had liver metastases [91]. Moreover, increased levels of SHAP-HA complex have been shown as a significant independent variable for progression-free survival in endometrial and ovarian cancer [95,134]. Increased LYVE-1 expression is associated with lymphatic metastasis and unfavourable prognosis in neuroblastoma [121]. Contrastingly, in lung cancer patients, reduced serum levels of LYVE-1 showed a significant association with lymph node metastases and distant metastases [125]. The number of LYVE-1-positive lymphatic vessels in ductal carcinoma of the breast was associated with reduced 5-year disease-free survival (DFS) [122]. Similarly, in endometrial carcinoma, elevated peritumoural expression of LYVE-1 was shown as an independent prognostic factor for OS and relapse-free survival (RFS) [123]. RHAMM expression was associated with poorer survival in large-cell carcinoma (LCC) [135]. HAPLN1 expression is upregulated in pancreatic cancer and malignant pleural mesothelioma [108], but its expression is lost in colorectal cancer (CRC) [110]. It has been shown as an independent risk factor for poor prognosis in CRC [136].

Increased expression of HAPLN3 has been reported in breast cancer [112] while TSG-6 has been found to be increased in colon cancer, ovarian cancer, bladder cancer, and prostate cancer [87,88,89] In colon and ovarian cancer, higher TSG-6 levels were correlated with poor prognosis [89]. Notably, TSG-6 is overexpressed in CRC, high-grade urothelial carcinomas, and high-grade prostate cancers [87]. Its expression level in CRC was associated with poor prognosis and invasive traits. TSG-6 may promote cancer metastasis through autocrine and paracrine pathways [88,137]. Furthermore, PHBP mRNA levels were found to be upregulated in non-small-cell lung cancer (NSCLC) [104], while a downregulation and loss of function mutation of HABP2 were seen in head and neck squamous cell carcinoma [105] and thyroid carcinoma [106], separately. 

Increased expression of lectican family members has also been revealed in tumours [127,128,129,131]. 

BCAN was reduced in NSCLC patients with shortened survival time and was used in a hypoxia-related scoring model to predict the prognosis of NSCLC [138]. However, higher BCAN levels might also be related to worse prognosis in lung adenocarcinoma (LUAD) [139]. NCAN mRNA was detected in a mouse model of breast cancer [140]. NCAN is significantly increased in astrocytoma, glioblastoma, and other tumours [128]. NCAN expression is strongly associated with poor prognosis in patients with neuroblastoma (NB) [129]. Low NCAN expression was associated with lower Merkel Cell Carcinoma (MCC)-specific survival compared with NCAN intermediate- and high-expression groups (*p* = 0.044) [141]. VCAN is widely expressed in the human body and imparts hygroscopic properties to the extracellular matrix (ECM), forming a loosely structured and hydrated matrix which is essential for supporting crucial processes in both development and disease [50]. Accumulation of VCAN in peritumoural stroma was associated with the prognosis of node-negative breast cancer [142], early stage prostate cancer [143], NSCLC [144], oral squamous cell carcinoma [145], testicular germ cell tumours [146], cervical cancer [147], endometrial cancer [148], and epithelial ovarian cancer [149]. High VCAN expression was reported to be associated with poor prognosis in gastric cancer (GC) patients [149].

HAIMs play important roles in disease progression. ITIH3 is reduced in colorectal, breast, uterine, ovarian, and lung cancers compared with adjacent healthy tissues [56]. ITIH3 expression was significantly higher in patients with advanced pancreatic ductal adenocarcinoma (PDAC) [93]. RHAMM overexpression is associated with the progression and metastasis of various tumours, including pancreatic, gastric, endometrial, breast, ovarian, colon, bladder, liver, and lung cancers [100,101,102,133,150,151,152,153,154,155]. Together with dysregulated p53, RHAMM overexpression is associated with worse clinical outcomes in ovarian cancer and pancreatic cancer [99,132]. Considering its detectability in urine and correlation with disease progression, RHAMM was proposed as a potential marker of ovarian cancer recurrence [99]. LYVE-1 has been extensively studied in cancer metastasis. LYVE-1 expression is increased in both GC and CRC [119,120]. Knockout of LYVE-1 in a mouse model resulted in an inhibition of liver metastasis of melanoma cells but had no effect on liver metastasis from a CRC cell line (MC38) [156]. In relation to tongue squamous cell carcinomas, a reduction in the density of LYVE-1-positive lymphatic vessels at the tumour invasion site and submucosal tissue around the tumour may serve as indicators of lymph nodes involvement [126]. Furthermore, LYVE-1 can be used to differentiate between benign and malignant vascular tumours, as all angiosarcomas and Kaposi’s sarcoma are positive for LYVE-1 [124]. An upregulation of HAPLN1 has been reported in pancreatic cancer and malignant pleural mesothelioma [108], while its expression is lost in CRC [110]. HAPLN1 expression is enriched in basal subtype PDAC, which is associated with poor OS and peritoneal metastasis [107]. An upregulation of HAPLN1 in cancer-associated fibroblasts (CAF) has been reported in GC, which is associated with an aggressive phenotype, poor prognosis, and tumour progression [109]. PHBP is activated as local capsular invasive carcinomas (CICs) lesions acquire a malignant phenotype and develop into carcinomas [157]. An elevated transcript level of PHBP was found in lung adenocarcinomas [103], and the upregulation of PHBP was also reported to be related to pathological status in non-small-cell lung cancer [70,104]. Both HABP2 and HABP4 seem to be tumour inhibitors [106,158]. Among the lecticans, BCAN overexpression is associated with glioma progression [159]. VCAN has been found in advanced non-seminomatous germ cell tumours [146]. 

The altered expression of these genes is also associated with chemoresistance. Interestingly, decreased ITIH2 expression levels were associated with bortezomib resistance in multiple myeloma [160]. HAPLN1 induced multi-drug resistance to bortezomib and ixazomib in patients with multiple myeloma [161]. GC patients with high VCAN are often resistant to immunotherapy [149]; in one study, cervical cancer patients who were resistant to chemotherapy also presented high expression of VCAN [162]. Similarly, elevated RHAMM expression in gliomas is correlated with worse clinical outcomes and higher risk of chemoresistance [163]. 

In addition to the aberrant expression level, mutations and alternative splicing variants of HAIMS are also involved in the disease progression of certain malignancies. Diffuse RHAMM expression significantly affects the survival time of CRC patients, making its expression level a more important prognostic indicator than tumour grade, tumour budding, and vascular invasion [164]. C-terminal deletion of RHAMM presents higher intracellular distribution than its wild-type format and, together with TP53 dysfunction, leads to lower survival in pancreatic cancer patients [132]. The HAPB2 G534E variant makes HABP2 lose its tumour inhibition function in familial nonmedullary thyroid cancer. Its protein expression was found to be upregulated in neoplasms compared to normal thyroid tissue [106]. Functional studies have revealed that PHBP variants also contribute to cardiovascular disease and thromboembolism [165,166]. The RHAMM spliced variant RHAMMB was more abundant in advanced PDAC that developed liver metastases. A high level of RHAMMB was also correlated with higher EGFR levels and poor clinical outcomes [98]. Five human VCAN splice variants (V0, V1, V2, V3, and V4) have been characterised [167]. The distinctions between these VCAN splicing variants reside in the central region of the protein core, exhibiting variations in the number and presence of glycosaminoglycans (GAGs). V0 and V1 are the main variants in cancers [168,169,170], while V3 variant seems to be a dual regulator in cancer progression due to its inhibiting effect on tumour growth and stimulating effect on metastasis in melanoma [171]. BCAN has several isoforms that function differently. B/b (Deltag) with minimal glycosylation was exclusively presented in glioblastoma multiforme (GBM) tissues and was a major isoform that was upregulated in high-grade glioma but was absent in the low-grade type, and its mechanism of binding to the cell membrane appeared to be different from that of other BCAN types. This isoform is targetable by a small peptide candidate [172]. NCAN rs2228603 polymorphism T allele was significantly higher in patients with hepatocellular carcinoma (HCC) due to alcoholic liver disease (ALD) and may serve as a risk factor [173]. NCAN in the ECM was also digested by matrix protease, which is helpful to CNS repair [174]. But whether this modification is related to solid tumour development is poorly understood.

Levels of post-translational modifications of genes, such as methylation, have also been found to be altered in prostate cancer [117]. DNA epigenetic analysis showed HAPLN3-methylated circulating tumour DNA(ctDNA) was widely found in de novo metastatic PCa (mPCa) and was markedly elevated in high-volume mPCa. Furthermore, the identification of methylated ctDNA was linked to a notably reduced period until the progression to metastatic castration-resistant PCa [118]. Combined with the signature of other genes, HAPLN3 methylation has been used as a biomarker to predict prostate cancer recurrence [175].

**Table 2 cancers-16-01907-t002:** Aberrant HAIM levels in cancers.

HA Interacting Molecule	Abnormal Expression		Mutation/Isoform
	Elevation	Decrease	
TSG-6	High-grade prostate tumour [87], colon cancer [88], ovarian cancer [89]		
ITIH2	Longer survival in LIHC and colorectal cancer liver metastasis [90,91]	Hepatocellular carcinoma (HCC) cells [92]	
ITIH3	Advanced PDAC [93]	Colorectal, breast, uterine, ovarian, and lung cancers [41,56]	
SHAP	Breast, ovarian, and endometrial cancer [94,95,96]		
RHAMM	Breast, ovarian, and pancreatic cancer; lung cancer, endometrial cancer, bladder cancer, hepatocellular carcinoma, and colon cancer [97,98,99,100,101,102,132,133,155]		Pancreatic cancer [132].
PHBP	Lung adenocarcinomas [103] and non-small-cell lung cancer [104]	Head and neck squamous cell carcinoma [105]	Thyroid cancer [106]
HAPLN1	Pancreas cancer [107], malignant pleural mesothelioma [108], lung cancer [109]	Colorectal cancer [110], malignant gliomas [111]	
HAPLN2		Malignant gliomas [111]	
HAPLN3	Breast cancer [112,113], clear cell renal cell cancer [114]	Advanced skin cutaneous melanoma [115], cutaneous melanoma [116]	Gene methylation: prostate cancer [117] and de novo metastatic prostate cancer [118]
HAPLN4		Malignant gliomas [111]	
LYVE-1	Breast, endometrial carcinoma, gastric cancer, malignant vascular tumours, neuroblastoma, and colorectal cancer [119,120,121,122,123,124]	Tongue squamous Cell carcinomas, lung cancer metastasis [125,126]	
NCAN	Astrocytoma, glioblastoma, neuroblastoma [127,128,129]		
BCAN	Glioma [50,159]		B/b(Deltag): only present in high-grade glioma [130]
VCAN	Brain tumours, melanomas, osteosarcomas, lymphomas, acute monocytic leukaemia, testicular tumours, breast, prostate, colon, lung, pancreatic, endometrial, ovarian, and oral cancers [131]		

## 4. HAIM Coordinated Cellular Functions

In addition to their interaction with HA, the deregulated HAIMs in cancer also play important roles in the regulation of cellular functions and plasticity of cancer cells to facilitate tumourigenesis, local invasive growth, and dissemination to distant sites (Figure 4 and Figure 5). 

Dedifferentiation and epithelial mesenchymal transition (EMT)

CD44, RHAMM, TSG-6, and NCAN have been reported to play a profound role in dedifferentiation during tumourigenesis and disease progression [129,176,177,178]. CD44 expression is increased in cancer cells undergoing EMT and those cancerous cells with acquired stem-like properties [176]. RHAMM can induce dedifferentiation, while inhibition of RHAMM can reverse the malignant phenotype of fibrosarcoma [177]. On the other hand, topical administration of HMW-HA in the treatment of enteritis accelerates intestinal epithelial regeneration in a TSG-6-dependent manner [178].

NCAN can promote dedifferentiation of neuroblastoma (NB) cells and can stimulate their malignancy [129]. PTPRσ may be a potential receptor for NCAN. Monomeric PTPRσ binds and activates lecticans, which in turn dephosphorylate downstream substrates, including receptor tyrosine kinases such as EGFR and FGFR1, to maintain the undifferentiated status of neurons [179,180]. In contrast, neuronal differentiation was found to increase in PTPRσ-deficient neurons [181]. Heparan sulphate proteoglycan (HSPG) facilitates the interaction between heparin-binding epidermal growth factor-like growth factor (HB-EGF) and EGFR, leading to the differentiation of NB cells via an activation of ERK1/2 and STAT3 pathways [182]. However, upon interacting with HSPGs, NCAN-PTPRσ-mediated regulation of EGFR and FGFR1 will be weakened and therefore will contribute to the undifferentiated states of NB cells [129].

Proliferation

Certain HAIMs, including CD44, TSG-6, LYVE-1, RHAMM, HAPLN3, NCAN, and VCAN, can facilitate the proliferation of cancer cells [57,85,100,114,155,183,184,185,186,187,188,189]. CD44 is essential for the rapid growth of various cancer cells [183]. TSG-6 helps maintain the proliferation ability of canine breast cancer cells under hypoxic conditions, which is related to the maintenance of G2/M phase [184]. TSG-6 contributes to cell survival under adverse conditions [184,185,186]. Both TSG-6-deficient mesenchymal stem/stromal cells (MSCs) and breast cancer cells showed a decreased proliferative rate [185,186], in which an upregulation of PD-L1 was observed [186]. A combination of LMW-HA and LYVE-1 promoted the proliferation and migration of lymphatic endothelial cells, thereby promoting lymphangiogenesis [187], while proliferation of melanoma cells is inhibited by ectodomain of LYVE-1 that was shed from M2-like tumour-associated macrophages. RHAMM regulates cell proliferation by coordinating cell cycling and cell division in a different manner from TSG-6 [100]. Unlike CD44 and LYVE-1, RHAMM exists in the cell membrane, cytoplasm, and nucleus. Extracellularly, RHAMM binds to CD44, HA, and growth factor receptors (GFR), and subsequently activates the ERK1/2 MAPK pathway to promote cell proliferation, migration, and invasion [57]. Intracellular RHAMM can enter the cell nucleus, activating ERK to regulate the movement of microtubules and centrosomes [57,190,191]. RHAMM also modulates host cell responses to affect the survival of tumour cells [155]. HA binding with CD44 can facilitate EGFR-promoted cell proliferation, in which RHAMM is required for the activation of Aurora Kinase A (AURKA) [66]. In various cancer models, HA amplifies cell proliferation and migration by engaging with RHAMM and activating additional receptors that govern the PI3K/AKT and ERK pathways, including CD44, PDGFR, and EGFR [163]. Normal human lung fibroblasts (NHLF)/lung cancer-associated fibroblasts (LCAF) promote proliferation of NSCLC cells through the HA-CD44/RHAMM signalling pathway in an HA-dependent manner [154]. However, the growth of malignant peripheral nerve sheath tumours is independent from RHAMM [57]. 

HAPLN3 acts as an oncogenic factor in cells, and a loss of HAPLN3 hinders the ability of clear cell renal cell carcinoma (ccRCC) cells to proliferate, migrate, and invade by suppressing activation of ERK1/2 [114]. NCAN is mainly expressed in the nervous system and works as an inhibitory molecule to inhibit axon regeneration following a nerve injury [51], but acts as a promoter for the proliferation of neuroblastoma cells [129]. VCAN has the capacity to enhance proliferation; the G3 domain of VCAN is essential to its promotive effect on proliferation, adhesion, and migration of endothelial cells (EC) [85]. The EGF-like motif within the G3 domain of VCAN enables its activation of EGFR and the function as a mitogen [192]. However, VCAN can promote apoptosis in breast cancer [86]. Interestingly, VCAN itself has also been reported to play an anti-apoptotic role like TSG-6 [188,189]. VCAN V1 exhibited an ability to regulate the tumour cells’ sensitivity to apoptosis [189]. Apart from the tumour itself, VCAN also regulates the proliferation of crucial components associated with tumours. For example, platelet-derived growth factor (PDGF) induces VCAN expression in arterial smooth muscle cells to facilitate the expansion of the pericellular extracellular matrix (ECM) during the tumour growth and dissemination of cancer cells [193]. PHBP promotes proliferation of fibroblasts [194] but its role in cancer cells remains largely unknown. More intensive staining of PHBP protein has been observed in thyroid lesions progressing from hyperplasia to carcinomas which implies a role of PHBP in the acquiring of a malignant phenotype in thyroid [157].

Adhesion

CD44, RHAMM, HAPLN4 and BCAN can enhance cell adhesion [111,159] while BCAN can also facilitate tumourigenesis [159]. An absence of CD44 can lead to a decrease of invasion and adhesion of breast cancer cells to ECs in vitro, while having no impact on cell proliferation [195]. Reducing RHAMM expression can inhibit cell adhesion to ECM while CD44 remains unaffected [196]. RhoA and RhoC can upregulate RHAMM via the activation of YAP [197]. When RHAMM expression is decreased by a Rho inhibitor Rhosin, cell adhesion to type I and IV collagen can be prevented [197]. RHAMM-mediated signalling may lead to conformational changes in integrins, reducing their capacity for cell adhesion [198]. In fibrosarcoma cells, LMW-HA amplified basal and adhesion-dependent phosphorylation of ERK1/2 and focal adhesion kinase (FAK) to promote adhesion in a manner dependent on RHAMM [199].

BCAN can enhance the activation of EGFR, elevate the expression of cell adhesion molecules, and stimulate the secretion and accumulation of fibronectin [200]. This facilitates the dispersion of glioma cells, but this only happens after a proteolytic cleavage of its N-terminal [200]. Similarly, cleavage of NCAN also enhances the adhesion of neuroglioma cell adhesion [174]. The G1 domain of VCAN bears its anti-adhesion ability [83,201]. Soluble VCAN can diminish attachment of prostate cancer and melanoma cells to surfaces coated with fibronectin [202,203].

Invasion and motility

CD44, RHAMM, TSG-6, HAPLNs, PHBP, BCAN, NCAN, and VCAN are able to promote cell invasion and motility, though some of these molecules may rely on the existence of other molecules or a cleavage. Cancer cells with CD44 overexpression present enhanced invasiveness and chemoresistance [204]. HA-CD44 binding can promote the invasion of metastatic colon cancer cells [25]. CD44 can also promote migration of malignant pleural mesothelioma cells [205]. In addition, the involvement of MMPs in ECM, CD44 further regulates cell invasion and angiogenesis [25,206,207,208]. Activation of MMP9 by CD44 expressed on the cell surface is associated with increased migration and invasion of prostate cancer cells (PC-3), which can be weakened by inhibition of Rho-GTPase using bisphosphonate [206]. The CD44-MMP9-promoted invasion and migration can also be disrupted by a knockdown of MMP9, which may result in a transformation of CD44s to CD44v6 and a less invasive phenotype [209]. However, CD44v6 in CRC stem cells can promote their migration and is associated with poor prognosis in which cytokines released from the TME such as HGF, OPN, and stromal-derived factor 1α (SDF-1) may play a role in the expression of CD44v6 and activation of the Wnt/β-catenin pathway may also involve [210]. Targeting CD44v6 can reduce the invasive capacity of endocrine-resistant breast cancer cells [211]. RHAMM promotes glioma cell migration [212], and the HA/RHAMM interaction can promote metastasis and invasion through an activation of the RHO-ROCK, PI3K/Akt and MEK/ERK pathways [57,65]. Loss of RHAMM can attenuate the formation of filopodia and cell migration and cleavage of FAK [213]. Silencing of RHAMM hindered the migration and invasion of CRC cells [100]. RHAMM oversees the dynamics of microtubules and the structure/function of centrosomes through the activation of ERK1/2/MAP kinase and plays a vital role in microtubule-mediated cell polarity and migration [57]. Furthermore, secreted RHAMM has the capability to bind with HA and, in conjunction with CD44, promotes the invasiveness of breast cancer cells [57].

Some function of TSG-6 may be related to its competitive binding to CD44 with TLR [80]; it enhances the invasiveness of colorectal cancer cells in a CD44-dependent manner and creates a favourable microenvironment for metastasis by reprogramming normal fibroblasts [137].

HAPLN3 enhances the migration and infiltration of ccRCC cells by activating the ERK1/2 pathway [114]. Both HAPLN3 and HAPLN4 can enhance the migration of glioma cells, while HAPLN4 can strengthen the mitogenic effect of BCAN [111]. PHBP can promote migration of fibroblasts [194]. Inhibition of PHBP in rat thyroid cancer demonstrated an inhibition of tumour invasion and angiogenesis [157]. 

The invasiveness of low-grade astrocytoma is primarily attributed to BCAN, NCAN, Tenascin-C, and VCAN [128]. Although both full-length BCAN and BCAN HA binding domain (HABD) can promote invasion of glioma cells in non-neurogenic ECM, it seems that only HABD remains functional, promoting the invasion in brain ECM [159,214]. Invasion in brain tumours may be enhanced by BCAN-cleaved disintegrin and metalloproteinase with thrombospondin motifs (ADAMTS) [200]. The promoting effect of NCAN on cell adhesion and motility can be hindered because of its cleavage by ADAMTS12 [174]. VCAN forms complexes with HA, CD44, and other molecules, leading to the augmentation of the extracellular matrix. This results in heightened viscosity and elasticity, rendering it remarkably adaptive to the requirements of cancer cells for both proliferation and migration. Experimental investigations have demonstrated that VCAN can enhance the motility of cancer cells [85,215,216] and metastasis [217]. Both the core protein domain and glycosaminoglycan side-chains of VCAN can promote cell motility and invasion [84,215]. This can be achieved via a binding with HA and its multiple highly negatively charged chondroitin sulphate (CS) side chains. Cell surface CD44 and subsequently assembled HA/VCAN aggregates can assist cancer cells in establishing polarised pericellular sheaths, thereby augmenting their motility [215]. 

Both RHAMM and NCAN have the capability to enhance tumourigenicity in vivo. RHAMM knockdown resulted in a reduced tumour size in CRC [100]. Exogenously expressed (i.e., overexpressed, recombinant protein, and conditioned media) NCAN induces adherent NB cells to form spheroids and develop tumour in vivo, in which both the chondroitin sulphate glycans and the core protein of NCAN are involved [129].

## 5. HAIMs and Extracellular Matrix Remodelling

TME encompasses the tumour cells, neighbouring normal/non-transformed cells, ECM, and secreted molecules. In various cancers, accumulated HA in the ECM is significantly related to the degree of malignancy [25,27,30,218]. This is jointly regulated by HAS and HYAL during the disease progression [16,218]. As is more extensively discussed in later sections, cancer cells, immune cells, CAFs, angiogenesis, and lymphangiogenesis in the TME are precisely regulated by the interaction between HA and HAIMs, thereby providing a favourable environment to facilitate tumour growth, invasion, and metastasis. However, the remodelling of ECM also plays a vital role in this process. 

The secreted HAIMs including IαIs, PHBP, TSG-6, HAPLNs and lectins are the key molecules that reshape ECM in the TME. By forming crosslinks with heavy chains, the IαI family plays an important role in stabilising HA chains and preventing the degradation by hyaluronidase [52]. It is noteworthy that apart from HA, HCs also interact with other ECM components, including vitronectin and fibronectin [52]. These interactions indicate that HCs may be an HA linker and may help to achieve more diverse biological functions. The HA/HC link is dynamic, and TSG-6 can reversibly shift HCs between HMW-HA chains and bikunin [52], therefore flexibly regulating ITIH and HA/HC complex density in the ECM to support different cellular functions [52]. PHBP exhibits similar functionality to TSG-6 and weakly cleaves the HC chain of IαI [219]. The regulation of PHBP by HA depends on its molecular size. HMW-HA in healthy tissues inhibits PHBP expression, while its proteolytic activity is activated by HA fragments under the inflammatory condition [70]. Its activation may be associated with ECM remodelling, leading to a promotion of cell invasion, and has been reported to correlate with a malignant phenotype [157].

HAPLNs and lecticans are the two protein families with link modules. Their locations in the mammalian genome are physically adjacent, and their functions are inseparable. HAPLNs are responsible for connecting lecticans to the HA chain, assisting in shaping ECM, especially in the brain ECM. It is noteworthy that HAPLNs’ binding to lecticans appears to be specific [220,221], and the lecticans’ binding with tenascin-R makes the HA scaffold more stable [221]. Similar to TSG-6, VCAN is another protein that assists the IαI-HC transfer to the HA chain and facilitates the binding of the SHAP to HA [222,223,224,225]. When VCAN forms complexes with molecules such as HA and CD44, the viscosity and elasticity of the ECM increase, shaping an environment conducive to cancer cell proliferation and migration. 

These HAIMs, by interacting with HA and even through HA’s interactions with other ECM components, alter the microstructure of the ECM, providing conditions for subsequent changes in cell behaviour regulation. Understanding their role in ECM remodelling is crucial for comprehending malignant cell behaviour.

## 6. HAIM and Angiogenesis/Lymphangiogenesis in Cancer

Angiogenesis and lymphangiogenesis are not only important events during the growth and expansion of primary tumours but also are pivotal for the spread of cancer cells to regional lymph nodes and distant sites. How VCAN, TSG-6, RHAMM, and CD44 regulate angiogenesis/lymphangiogenesis is illustrated in Figure 6.

A positive relationship between CD44 and micro-vessel density (MVD) has been evident in certain cancers [226,227,228]. CD44 expressed by ECs can promote angiogenesis, while an inhibition of CD44 warrants generation and stabilisation of endothelial tubular network [229]. Enhanced CD44 expression was found in solid tumour ECs compared to normal tissue [230]. Pro-angiogenic factors promote CD44 expression on tumour vasculature ECs [230]. The increase in these adhesion molecules on ECs not only promotes leukocyte adhesion and extravasation but also facilitates their adhesion to ECM components and generates new blood vessels [230]. Among the variants, CD44v6 plays an essential role in tumour-associated angiogenesis [230]. Upon binding with immobilised HA, HA-CD44 can promote angiogenesis by directing the lamellipodial protrusions formation of vascular ECs [207]. MMPs, including MT1-MMP, MMP2, and MMP9, are also involved in the CD44-promoted formation of new vasculature and tumour invasion, which can be enhanced by an activation of TGF-β with proteolytically activated MMP-9 [207]. 

HA appears to exert its angiogenic effects by binding and interacting with some HAIMs, including CD44 and RHAMM, but through different mechanisms [231]. CD44 facilitates endothelial cell proliferation and adhesion, while RHAMM not only promotes invasion of ECs through the basement membrane but also enhances basic fibroblast growth factor-induced angiogenesis [231]. Mechanically, the pro-angiogenic effect of HA may be achieved through CD44-PKCδ inducing RHAMM-TGFβ receptor interaction to induce the expression of plasminogen activator inhibitor-1 (PAI) [232]. Notably, the HA produced by tumour cells can promote both invasion and angiogenesis through an interaction with RHAMM and other HAIMs on the surface of ECs [233]. Contrastingly, TSG-6 is required for MSCs to inhibit angiogenesis and lymph-angiogenesis [234]. TSG-6 can inhibit FGF2-induced angiogenesis by competitively binding with pentraxin 3 (PTX3) [235]. HMW-HA helps TSG-6 to promote epithelial cell regeneration during wound healing [178], while HMW-HA-TSG-6 complex modulates the inhibition of angiogenesis in the context of breast cancer by downregulating TSG-6 levels in macrophages/monocytes [236]. Additionally, LMW-HA activates PHBP proteinase activity and impaired the vascular integrity [70]. This regulation is achieved via a cascade of activation of protease-activated receptor (PAR)/RhoA/ROCK kinase signal pathway [70].

Since ITIHs are essential for maintaining HA junction and ECM stability, dysregulation of ITIH family members may affect vascularisation during tumour development. In endometrial cancer, SHAP-HA complexes promote the involvement of the lymph vessels and the synthesis and activation of MMP-9 and TIMP-1, leading to the disease progression. SHAP-HA is therefore considered to be a useful marker for predicting the recurrence of endometrial cancer [95].

VCAN is also positively correlated with MVD, and its G3 domain is critical for VCAN-promoted angiogenesis both in vitro and in vivo [85,146,217]. VCAN can be synthesised by cancer cells, ECs, and adjacent fibroblast cells, leading to an accumulation of VCAN in the ECM during the angiogenic process [85]. Mechanically, VCAN directly interacts with fibronectin, which subsequently forms a complex with VEGF to enhance the adhesion, proliferation, and migration of vascular ECs, and therefore promote angiogenesis [237]. In breast cancer, the VCAN G3 domain interacts with ECM components, including fibronectin and VEGF, to promote angiogenesis [217]. Additionally, elevated VCAN V2 variants in glioma cells can promote new vasculature formation [237]. VCAN’s pro-angiogenic effect is also a result of its interaction with HA, fibronectin, and VEGF in the ECM [217,238]. The VCAN-HA complex can regulate the vascular and perivascular elastic structures of malignant tumours and enhance stromal cell recruitment to facilitate endothelial cell infiltration [238]. Therefore, the HAIMs play a pivotal role in angiogenesis, with therapeutic potential.

The lymphatic system is an extensive tubular network that transports white blood cells through the lymph nodes and the thoracic duct back into the blood. This network is involved in the transport and recycling of extracellular matrix components such as HA and plays an important role in immune monitoring. However, they also provide pathways for the dissemination of tumour cells to lymph nodes. LYVE-1 is homologous to CD44 and has a similar structure, but it only exists on the endothelial surface and lumen of lymphatic vessels and is absent in blood vessels [45]. As a distinct indicator of lymphangiogenesis, it also fosters the formation of new lymphatic vessels. HA/LYVE-1 binding regulates cell adhesion, the entry and exit of lymphatic vessels, and activated intracellular signal cascades to enhance lymphangiogenesis and tumour lymphatic metastasis [45,60,187,239]. Interestingly, LYVE-1-positive lymphatic vessels appear to be present only on the periphery of endometrial and lung cancers, but not in solid tumours [240]. The ectodomain shedding of LYVE-1 induced by VEGF-A can stimulate pathological lymphangiogenesis. The shedding process can be facilitated through ERK and a disintegrin and metalloproteinase (ADAM) 17 [241]. However, when the LYVE-1 extracellular domain on lymphatic ECs is cleaved by membrane type 1-matrix metalloproteinase (MT1-MMP), the lymphagiogenesis is consequently inhibited [242]. In addition, Nectin-4 regulated CXCR4/CXCL12-LYVE-1 axis plays an important role in both lymphangiogenesis and lymphatic metastasis in breast cancer [243]. Nevertheless, lymphangiogenesis is hindered by (MT1-MMP), which acts as a natural suppressor of lymphatic vessel development by directly cleaving LYVE-1 [242].

## 7. HAIM and Immunity in Cancer

Phagocytosis, inflammatory responses, immune organs, and cells play crucial roles as components of the immune system in the defence of the human body. Inflammation is frequently linked to the initiation and progression of cancer. HAIMs, together with inflammation-associated cells—particularly tumour-associated macrophages—contribute to angiogenesis, ECM degradation and remodelling, and cancer cell invasion. How these HAIMs interact with non-tumour cells and reshape ECM during the immune response is illustrated in Figure 7. 

Lymph nodes, as crucial components of the immune system, house a category of lymphocytes, including B cells and T cells. LYVE-1 is linked to the development of lymph nodes and emergence of an adaptive immunity [244]. The LYVE-1 expression pattern implies the transfer of HA from tissues into the lymphatic vessel, where it is internalised and degraded and provides a chemotactic substrate for CD44+ leukocytes and cancer cells [61]. LYVE-1 binds to HA, mediates immune cell trafficking within lymphatic vessels, and initiates immune responses [245]. The LYVE-HA axis mediates trafficking of dendritic cells (DC), macrophages and other lymphocytes in lymphatic vessels, thus initiating an immune response [245]. 

TSG-6 mediates immunosuppressive effects on the innate immune system, and its substantial upregulation in response to inflammation plays a pivotal role in various biological processes, including inflammation, wound healing, ovulation, and organ morphogenesis [55,73,246]. Under inflammatory conditions, TSG-6 exerts anti-inflammatory and tumour-inhibiting activities by inhibiting granulocyte migration and maintaining mesenchymal cell stemness [185,247,248]. Responding to tissue damage, TSG-6 produced by activated monocyte-derived macrophages (MØ) can be elevated by the inflammatory factors interleukin-1 (IL-1), IL-6, and TNF. The upregulated TSG-6, in turn, downregulates these molecules and causes ECM degradation, thereby exerting anti-inflammatory activity in which NF-κB and PI3K/Akt pathways are involved [55,249]. TSG-6 is a potent inhibitor of neutrophil migration, and it is crucial for maintaining the stemness and biological properties of mesenchymal stem cells (MSCs) [185]. It can also attenuate inflammatory responses by inhibiting neutrophil migration across the ECs, potentially through a direct interaction with the chemokine CXCL8 [248]. Notably, MSCs lacking TSG-6 release more IL-6, conferring pro-inflammatory and pro-tumour properties [185]. The SHAP-HA complex formed with the help of TSG-6 is increased in the pathological synovial fluid of arthritis patients, which is associated with the degree of inflammation, indicating a pivotal role of SHAP-HA in the inflammatory response [250]. This has also been observed in inflamed tissue from patients with inflammatory bowel disease [251] and in a canine tooth infection experiment [252].

RHAMM, expressed in activated T cells, is an immunogenic antigen associated with inflammation, fibrosis, acute and chronic leukaemia, breast cancer, and prostate cancer, and plays an important role in the immune system by regulating immune cell movement [133,253]. Overexpression of RHAMM by activated T cells induces both humoral and cellular immune responses, and vaccines targeting RHAMM can effectively inhibit cancer progression [254,255]. RHAMM expressed on the cell surface promotes the motile behaviour of progenitor thymocytes (CD3-CD4-CD8-) and malignant B cells from myeloma and leukaemia, while the soluble RHAMM impedes the locomotion of fibroblasts [256]. In mice with melanoma, a Xenopus receptor-based RHAMM (xRHAMM) DNA vaccine is capable of eliciting antigen-specific cellular responses and humoral immune reactions against RHAMM. This vaccine can also inhibit tumour-associated angiogenesis and promote cell apoptosis, ultimately suppressing tumours and metastasis to lungs [255]. Increased expression of HAS, CD44, and RHAMM creates a favourable condition for immune cell infiltration and cell proliferation in breast cancer [257]. Inhibiting RHAMM reduces inflammation and fibrosis by coordinating recruitment of macrophages and fibroblasts, and cytokine expression [67]. 

The infiltration of immune cells into tumours has a profound impact on the outcomes of cancer patients undergoing radiation, chemotherapy, and immunotherapy [258,259]. HAPLNs play a crucial role in regulating immune checkpoints and immune cell behaviour. Treating aged fibroblasts with HAPLN1 reinstates the mobility of mononuclear immune cells while hindering that of polymorphonuclear immune cells, subsequently affecting the recruitment of regulatory T-cells [260]. In ccRCC, HAPLN3 was positively associated with immune checkpoints, and a remarkable association was found between HAPLN3 and the key immune cells, including CD8+ T lymphocytes, CD4+ T lymphocytes, macrophages, neutrophils, and dendritic cells [114]. Through these mechanisms, HAPLN3 fosters the progression of ccRCC and may serve as a potential therapeutic target [114]. HAPLN3 has been proposed as a marker to predict survival and optimise the treatment of cutaneous melanoma [261]. Several pyroptosis-associated genes including HAPLN3 were highly associated with immune infiltration, immune checkpoints, treatment responses, immunoinflammatory response which have been used to evaluate immunity in the TME of cutaneous melanoma [262]. 

The role of VCAN in immune regulation is well studied, while the current research on other lecticans is limited, necessitating further investigation to elucidate their functions and underlying mechanisms. VCAN has the potential to foster the establishment of an inflammatory microenvironment within the tumour stroma. VCAN, along with pro-inflammatory cytokines, collaboratively promotes the establishment of an inflammatory environment in the TME [263]. The interaction between VCAN and TLR2 has been implicated in inflammation and metastasis [264,265]. Mechanically, VCAN activates ECs and fibroblasts by activating TLR2, leading to a release of inflammatory cytokines, including tumour necrosis factor-alpha (TNFα), IL-6, and other proinflammatory cytokines, which promote neutrophil infiltration, angiogenesis, and cancer metastasis [264,265,266,267]. Additionally, through a competitive binding to HA, VCAN inhibits the immune reactions activated by CD44 and suppresses immune responses by inhibiting T-cell migration [268]. The VCAN-HA complex provides a scaffold for leukocyte adhesion [269]. Fragments of this complex degraded by leukocytes exhibit pro-inflammatory characteristics, reinforcing the inflammatory response [269]. The involvement of VCAN in regulating ECM during the immune and inflammation process was reviewed by Wight and his colleagues [263]. Moreover, the inhibition of VCAN amplifies the anti-tumour effectiveness of endostatin by mitigating the induced accumulation of myeloid-derived suppressor cells (MDSCs), tumour-associated macrophages (TAMs), and inflammatory cytokines within the TME [270]. VCAN can also produce immunosuppressive effects by preventing the migration and invasion of T cells [268].

A recent study reported that BCAN, as a component in the basement membrane, possesses predictive potential for prognosis in the lung adenocarcinoma [139]. Additionally, secreted BCAN can be upregulated in astrocytes in response to CNS injury [271] which can be prevented in autoimmune encephalomyelitis mice in comparison with sham-immunised controls [272] due to a disturbed metabolism of GAGs. The function of BCAN in immune regulation remains controversial.

It is worth noting that, in addition to their individual roles, the binding of these proteins with HA also plays a crucial role in regulating immune responses to inflammatory stimulation. TSG-6 acts as a cofactor and catalyst in linking HC of IαI to HA, constructing a favourable ECM for cell expansion during the inflammation [73,80]. Upon an inflammatory stimulation, SHAP is recruited to the exterior of blood vessels, forming the SHAP-HA complex which interacts with inflammatory cells in the ECM. HMW-HA/SHAP also induces macrophages to adopt the anti-inflammatory M2 phenotype to prevent inflammatory responses [52]. Studies indicate that SHAP can alter the distribution of CD44 on the cell surface and increase the binding affinity of HA to CD44. Intriguingly, CD44-positive cells exhibit a preferential adhesion to SHAP-HA over HA, leading to a 20–30 times increase in CD44-mediated leukocyte adhesion [251,273].

The accumulation of VCAN in inflammatory tissues is often associated with HA, TSG-6, CD44, and IαI. These interactions collectively contribute to the formation of a niche with orchestrated inflammation and immune responses. Apart from TSG-6, VCAN also bind to HCs of the IαI family and assists in the binding of SHAP to HA by attaching SHAP to GAG chains; however, such a promotive effect can be alleviated by MMP3-mediated degradation of VCAN [252,274]. With a competitive binding, VCAN can interfere with HA-CD44 interaction in immune cells to weaken the immune response [269]. Moreover, VCAN and HA also serve as a scaffold to regulate leukocyte adhesion and activation both across and beyond vascular structures [269]. Leukocytes can degrade ECM via VCAN, generating proinflammatory fragments to intensify the inflammatory response by increasing the secretion of monocyte/macrophage-dependent proteases and proinflammatory cytokines [269]. 

## 8. HAIMs in Distant Metastasis of Cancer Cells

Distant metastasis is a complex and challenging journey, during which cancer cells colonise a distant site comprising orchestrated activities and events in which cancer cells orientated TME plays a pivotal role. In addition to the well-documented CD44, RHAMM, TSG-6, HAPLN, HABP4, and VCAN also play a profound role in distant metastasis, while LYVE-1 is mainly involved in lymph node metastasis. 

RHAMM performs multiple extracellular and intracellular functions related to metastasis, especially hematogenous metastasis, including regulating mitosis [275], enhancing genome stability [46], promoting cell motility and plasticity [276,277], facilitating the migration of vascular ECs and promoting angiogenesis [232,278,279], regulating pluripotency of stem cells [280,281] and activating oncogenic pathways [155]. Increased RHAMM expression in certain subpopulations of tumour cells is associated with distant metastases in breast cancer [133]. Blocking RHAMM can inhibit the development and spread of CRC [100]. Similarly, TSG-6 promotes the metastasis of CRC through both autocrine and paracrine pathways, in which CD44 is also involved [137]. Moreover, knockdown of RHAMM inhibits tumour growth and metastases in vivo [100]. Mechanistically, the interaction with HA and RHAMM modulates EMT, cell migration, and metastasis through a regulation of TGF-β1 and RHO-ROCK pathways [65]. 

HAPLN1 is the most-studied HAPLN family member in terms of metastasis. Its expression is enriched in basal PDAC subtypes and is associated with poor OS [107]. Upregulated HAPLN promotes peritoneal dissemination [107] and elevated expression of HAPLN1 in CAFs is associated with an aggressive phenotype, poor prognosis, and tumour progression in GC [109]. HAPLN1 can increase the permeability of lymphatic vessels by downregulating VE-cadherin in lymphatic ECs, which may contribute to in-transit metastases in melanoma, but this is associated with an increased risk of visceral metastases [282]. A loss of HAPLN1 in aging fibroblasts leads to structural changes in ECM and promotes melanoma cell migration [260]. A possible reason for this is that the small pores created by the highly crosslinked matrix are expanded in the abnormal ECM, which enables cancer cells to move through it. Understanding the aging-related changes of ECM in melanoma and the response to immune checkpoint inhibitors may help to improve the efficacy of immunotherapy. HAPLN1 is also associated with peritoneal metastasis in the pancreatic cancer [107], but its expression can inhibit lymph node metastasis in melanoma [282], suggesting context-specific functionality. Apart from HAPLN1, clinical data analysis showed that HAPLN3 overexpression was related to breast cancer metastasis [112] and its level was closely related to the depth of tumour invasion, lymph node invasion and distant metastases in ccRCC [114]. The latter correlation may be achieved by infiltration of tumour immune cells and evasion of immune checkpoints [114].

Regarding well-studied lecticans, involvement of VCAN in cancer metastasis has been reported in prostate cancer, testicular germ cell tumours (GCTs), breast cancer, and lung cancer. VCAN has been associated with the metastasis potential of malignant cells in prostate cancer [215] and GCTs [146]. In comparison with the primary tumours, increased expression of VCAN in the stromal tissue was seen in metastatic tumours [283]. Furthermore, VCAN released from bone marrow cells facilitates the metastasis of breast cancer, while VCAN expressed by CD11b + Ly6C-high cells induces lung metastasis through an EMT-dependent mechanism [284]. In the case of Lewis lung cancer (LLC), the VCAN produced by tumour cells may interact with the TLR2/TLR6 complex to promote metastasis of the lung, liver, and adrenal glands [265]. 

Lymph node metastasis refers to infiltrating tumour cells crossing the lymphatic vessel wall and subsequently being carried by lymphatic fluid to the convergence area in lymph nodes, forming new metastatic foci. In addition to HAPLN1, LYVE-1 is another key player in lymph node metastasis. Elevated LYVE-1 expression and the occurrence of lymphatic invasion are associated with the presence of lymph node metastasis in neuroblastoma [121]. LYVE-1 also plays a positive role in the lymph node metastasis of gastric cancer [120]. Immunohistochemistry for LYVE-1 can be used to detect lymphatic vessel invasion, which is an effective predictor of lymphatic metastasis in breast cancer [285]. A notable correlation was found between low LYVE-1-positive vessel density in preinvasive and peritumoural submucosal regions in tongue squamous cell carcinomas (TSCC) and the occurrence of regional lymph node metastasis [126]. 

## 9. Conclusions and Prospective

HAIMs play diverse roles by interacting with HA and reshaping the TME during cancer progression. In general, their function is closely related to their subcellular distribution. Cancer cells, immune cells, mesenchymal cells, and various ECM components in the TME precisely regulate the synthesis and degradation of HA. The interaction between HAIMs and HA in turn affects the immune response, angiogenesis, and malignancy of tumour cells in the TME in a coherent and orchestrated fashion. The expression of some HAIMs is tissue-specific, as there is a lack of extensive and in-depth reports regarding their role in solid tumours. 

Nanoparticles targeting RHAMM isoform B significantly reduced the tumour burden in a mouse model of pancreatic neuroendocrine tumour [286]. Short peptides with the HA-binding motif BX_7_B in RHAMM and CD44 have been shown to inhibit melanoma growth and induce apoptosis [287]. Other RHAMM mimic peptides have exhibited an inhibition of breast cancer cell invasion [288] and they can also induce prostate cancer cell apoptosis [289]. Studies also showed that vaccinating patients with RHAMM substantially diminished the tumour burden and enhanced the infiltration of RHAMM-specific cytotoxic T lymphocytes [255,290]. Subsequently, the effects of RHAMM-R3 peptide vaccine were tested in Phase I and Phase II clinical trials [291,292]. Administration of the RHAMM-R3 vaccine with chemotherapy or stem cell transplantation in patients with leukaemia can potentially delay relapse and maintain remission [292]. However, the multifunction of RHAMM and its specific expression in tissues have limited the application of these peptides. VCAN has been identified as a biomarker for tumour metastasis and chemotherapy resistance [293,294,295,296], assisting in the monitoring of off-target effects of platinum (Pt)-based chemotherapy and facilitating timely adjustments to therapeutic strategies [294]. Furthermore, VCAN is related to the efficacy of adjuvant chemotherapy, chemoradiotherapy, and immunotherapy in gastric cancer [149]. Targeting VCAN holds promise for enhancing the outcomes of current clinical treatments for gastric cancer.

However, the research on other molecules is limited due to their specific expression characteristics and low abundance in most tissues. Research related to LYVE-1 and cancer has primarily focused on its role in the lymphatic system and the immune responses it participates. However, recent studies have provided increasing indication that LYVE-1+ macrophages also play a significant role within the TME [297,298,299]. The currently available anti-LYVE-1 antibodies allow for in vivo fluorescence imaging of lymphatic vessels and the disseminating tumour cells [300], and are used for positron emission tomography (PET) scanning [301].

NCAN can potentially serve as a target for glioma therapy and a prognostic marker [302,303]. Increased NCAN expression is associated with improved survival rates in Merkel cell carcinoma (MCC), though the specific mechanisms are yet to be elucidated [141]. Interestingly, NCAN-GFP fusion protein can specifically bind and illustrate the distribution patterns of HA in tissue sections and even in live cells. 

Taken together, this review underscores the reciprocal regulation between HAIMs and HA, providing a new perspective for a deeper understanding of molecular interactions in the TME. It is crucial for unveiling mechanisms of tumour development, discovering novel therapeutic targets, and designing more effective anticancer strategies.

## Figures and Tables

**Figure 1 cancers-16-01907-f001:**
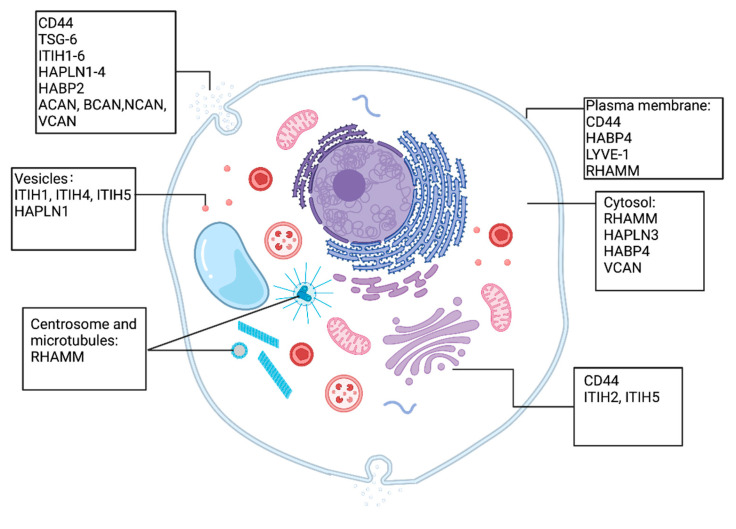
Subcellular location of HAIMs. Differences in the distribution of the HAIMs indicate a wide range of biological functions affected by these molecules. The figure was created with BioRender (www.Biorender.com, accessed on 8 May 2024).

**Figure 2 cancers-16-01907-f002:**
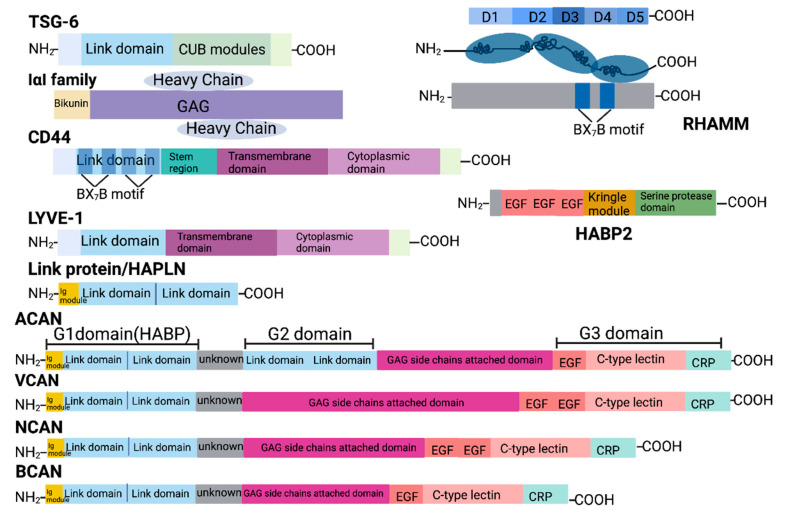
Protein structure of HAIMs. Domain architecture of HAIMs. The link domain on the N-terminal of TSG-6, CD44, LYVE-1, HAPLNs and lecticans is the HA-binding domain, while in RHAMM, BX7B motif helps to bind HA. The G1 domain in the lecticans is responsible for HA binding; the G3 domain binds to ECM molecules like Tenascin and carbohydrate, including GAGs on the cell membrane. G2 domain is only found in ACAN, but its function remains unknown. CUB = C1r/C1s, Uegf, Bmp1; Within the BX7B motif, B = arginine (R) or lysine (K); X = non-acidic amino; GAG = glycosaminoglycans; EGF = epidermal growth factor (EGF)-like motif; CRP = complement regulatory protein repeat. This figure was created with www.BioRender.com (accessed on 8 May 2024).

**Figure 3 cancers-16-01907-f003:**
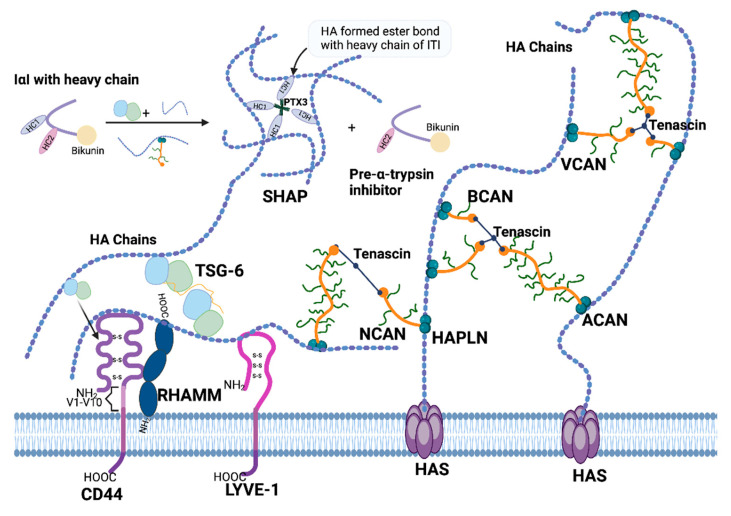
Interactions between HAIMs and HA in the ECM. HAIMs interact with HA to organise ECM. HA is produced directly by HAS. TSG-6 and VCAN assist with the transport of HC from IαI family members to HA chains. HCs linked to HA chains via ester bond and Pre-α-trypsin inhibitor are released. PTX3 helps with further HA organisation in the mammalian oocytes complex matrix. TSG-6 itself interacts with CD44 and enables CD44 to form a complex with HA. The induction of dimerization in TSG-6 by HA leads to the crosslinking of the HA polysaccharide. HAPLNs binding on the G1 domain of lecticans and tenascin’s binding to lecticans on their G3 domain also help to construct HA scaffold. PTX3 = Pentraxin-3; HC = heavy chain. This figure was created with BioRender (www.Biorender.com, accessed on 8 May 2024).

**Figure 4 cancers-16-01907-f004:**
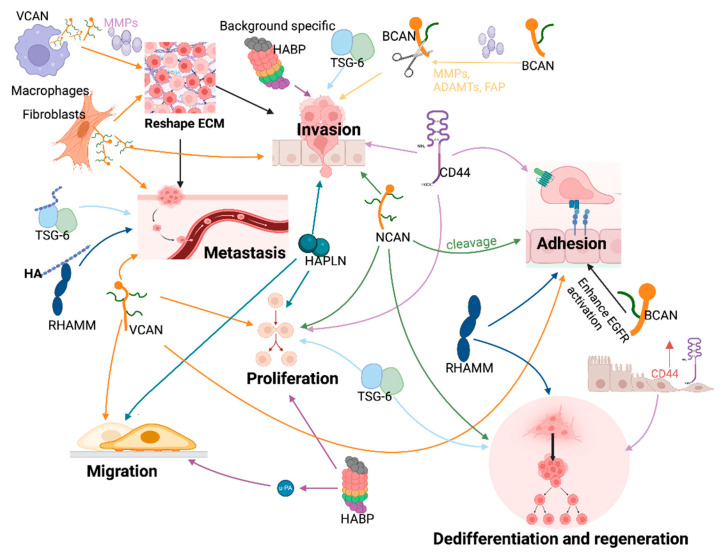
HAIMs modulate cellular functions and the dissemination of cancer cells by interacting with ECM components. HAIMs from cancer cells, immune cells and mesenchymal cells all contributed to the abnormal HAIM levels in tumours. These HAIMs either function alone or form complexes with other HAIMs and ECM components including HA to alter malignant cellular behaviours. Cleavage of lecticans will affect their function. Various colours are used for indicating different molecules. Arrows are used to show their promoting effects. MMP = matrix metalloproteinase; ADAMTS = A disintegrin and metalloproteinase with thrombospondin motifs; FAP = fibroblast activation protein; EGFR = epidermal growth factor receptor. The figure was prepared with BioRender (www.Biorender.com, accessed on 8 May 2024).

**Figure 5 cancers-16-01907-f005:**
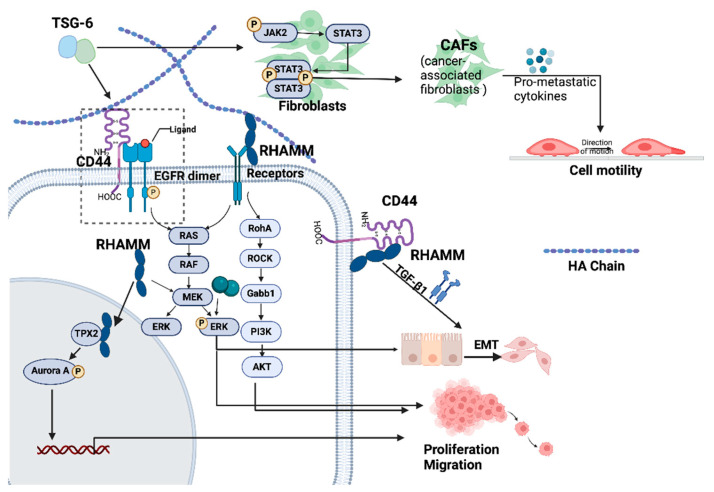
The molecular machinery of CD44, TSG-6, HAPLN and RHAMM regulated tumour proliferation, movement and metastasis. TSG-6 can facilitate CD44-EGFR interaction. HA/CD44 complex enhances EGFR-promoted cell proliferation and EMT via the RAS/TAF/MEK/ERK pathway. Membrane-anchored RHAMM–HA interaction activates RHO/ROCK pathway and therefore promoted cell migration and proliferation via unknown transmembrane receptors. Intercellular RHAMM enter the nucleus and binds to and stabilizes TPX2 to activate AURKA, leading to enhanced proliferation and migration. TSG-6 converts normal fibroblasts to CAF and therefore promotes CRC metastasis. RHAMM/CD44 complex also promotes EMT with the involvement of TGF-β1. AUKA = Aurora kinase A; TGFβ = transforming growth factor beta; EMT = epithelial–mesenchymal transition; EGFR = epidermal growth factor receptor. Created with BioRender (www.Biorender.com, accessed on 8 May 2024).

**Figure 6 cancers-16-01907-f006:**
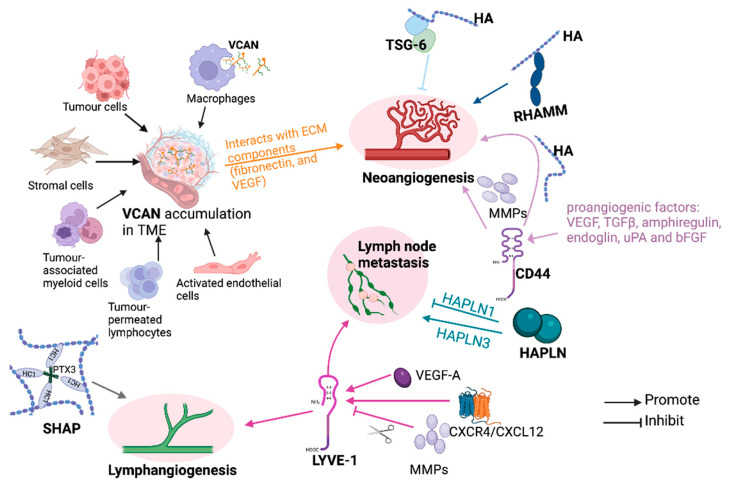
HAIMs’ involvement in angiogenesis and lymphangiogenesis. VCAN, TSG-6, RHAMM and CD44 contribute to angiogenesis while SHAP, LYVE-1 and HAPLNs are involved in lymph angiogenesis and lymph node metastasis. Various colours are used for the purpose of grouping, indicating distinct molecular functions. VEGF = vascular endothelial growth factor; TGFβ = transforming growth factor beta; uPA = urokinase-type plasminogen activator; bFGF = basic fibroblast growth factor; CXCR4/CXCL12 = chemokine (C-X-C motif) receptor 4/chemokine (C-X-C motif) ligand 12. Created with BioRender (www.Biorender.com, accessed on 8 May 2024).

**Figure 7 cancers-16-01907-f007:**
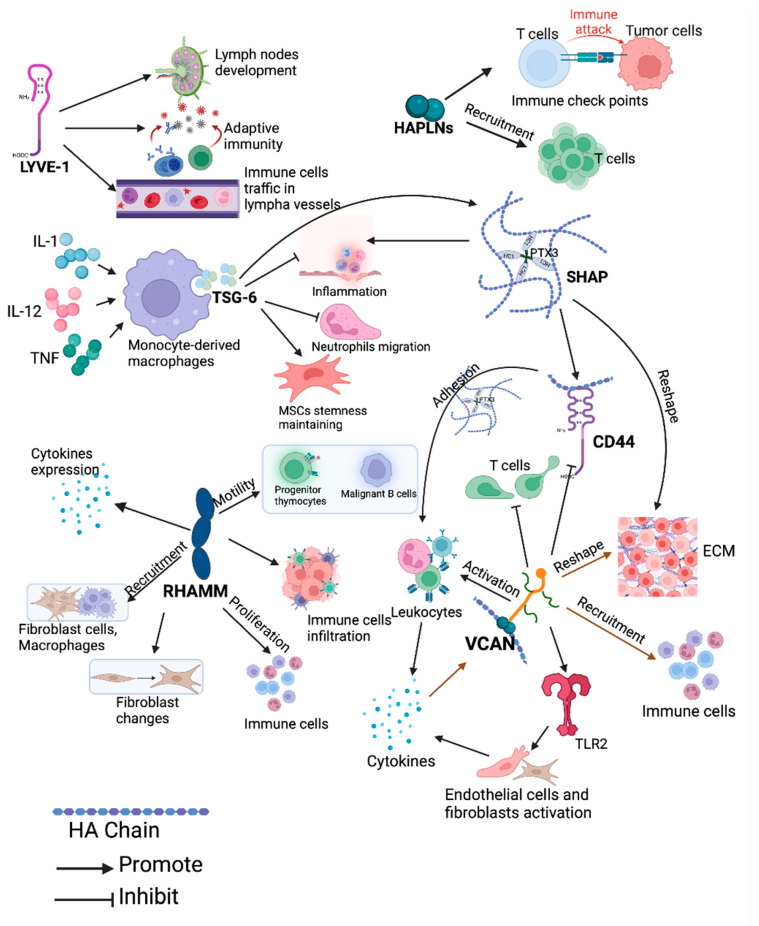
HAIMs’ involvement in immune regulation. Various colours are used for the purpose of grouping, indicating distinct molecular functions. HC = heavy chain; PTX-3 = pentraxin-3; IL = interleukin; TNF = tumour necrosis factor; TLR = toll-like receptor. Created with BioRender (www.Biorender.com, accessed on 8 May 2024).

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
