# Peer review of "Hyaluronic Acid Interacting Molecules Mediated Crosstalk between Cancer Cells and Microenvironment from Primary Tumour to Distant Metastasis"

_cancers, 2024, doi:10.3390/cancers16101907_

Round 1

Reviewer 1 Report

Comments and Suggestions for Authors

This review focuses on the impact of HAIMs on cancer progression and aims to highlight the multifaceted role of HAIMs in the context of cancer. Overall, this is informative and valuable for understanding the interplay between HAIMs and HA in the tumour microenvironment. However, I would appreciate your consideration of the following modifications to improve the integrity of manuscript.

1.     In Fig. 2, GAG represents both GAG and GAG attachment domain. To avoid any confusion, the authors need to distinguish between GAG in IαI and GAG attachment domains in CD44, ACAN, VCAN, NCAN, and BCAN.

2.     Hyaluronan displaces the chondroitin sulfate of bikunin and forms an ester bond with a heavy chain of ITI. It is recommended that details of this reaction will be added to Figure 3.

3.     The authors defined HMW-HA as >500 kDa and LMW-HA as <500 kDa. However, the function of HA is more tightly regulated by its molecular size. For example, HA of 10-250 kDa and HA oligosaccharides <10 kDa differ in their function. Please accurately describe the relationship between molecular weight and function of HA.

4.     Since Section 3 (Uncontrolled HAIM in Solid Tumors) is too long, this section should be divided into at least two sections.

5.     Given the overall structure of this manuscript, it would be appropriate for section 8 to be before section 5.

6.     In Fig. 2, dark purple should not be used because characters are not visible.

7.     For ITIH, ACAN, VCAN, NCAN, and BCAN, the authors should explain each abbreviation when it first appears in the main text. On the other hand, for Neurocan, Aggrecan, and Versican, the abbreviations should be listed on page 8, line 236. Similarly, on page 11, lanes 397-399, the authors should explain ctDNA when it first appears in the main text.

8.     On page 5, line 135 and page 20, line 755, IAI and IaI should be IαI.

9.     Please correct V3and V4 on page 10, line 380 and ECscan on page 16, line 570.

10.  For reference 1, a more appropriate citation is recommended.

Comments on the Quality of English Language

Minor errors such as capitalization, spaces, superscripts, etc. need to be corrected.

Author Response

Response to Suggestion1 and 6:

  1. In Fig. 2, GAG represents both GAG and GAG attachment domain. To avoid any confusion, the authors need to distinguish between GAG in IαI and GAG attachment domains in CD44, ACAN, VCAN, NCAN, and BCAN.

Authors: To avoid misunderstanding, GAG in ACAN, VCAN, BCAN and NCAN structure diagrams have been changed to GAG side chains. The GAG attachment of CD44 is added during post-translational modification. Since it is inconvenient to display GAG attachment in the structural diagram, we changed the Region to a Stem region that only exists in CD44 variants and changes with variant types. The GAG attachment of CD44 is added during post-translational modification. Since it is not convenient to display GAG attachment in the structural diagram, we changed the Region to a Stem region that only exists in CD44 variants and changes with the variant type. Reviews have shown that CD44 glycosylation has a bidirectional regulatory effect on its activation[1].

  1. In Fig. 2, dark purple should not be used because characters are not visible.

Authors: The dark colour had been replaced with lighter colour.

Response to Suggestion 2: Hyaluronan displaces the chondroitin sulfate of bikunin and forms an ester bond with a heavy chain of ITI. It is recommended that details of this reaction will be added to Figure

Authors: Already shown the process. We added a comment to explain that the connection bond between HA and HCs is an ester bond.

Response to Suggestion 3: The authors defined HMW-HA as >500 kDa and LMW-HA as <500 kDa. However, the function of HA is more tightly regulated by its molecular size. For example, HA of 10-250 kDa and HA oligosaccharides <10 kDa differ in their function. Please accurately describe the relationship between molecular weight and function of HA.

“Nevertheless, High-molecular-weight HA (HMW-HA) and low-molecular-weight HA (LMW-HA), including HA fragments, have distinct roles in the human body. The role of different size HA had been reviewed by Robert and his colleague [13]. HMW-HA (>250kDa) is a space-filling molecule with hydration capacity to retain water and create a supporting framework for cells. HMW-HA also functions as a lubricant and a liquid shock absorber in the articular cavity. Furthermore, it can also regulate intracellular signal transduction upon binding to cell membrane receptors like CD44 [14], thereby affecting various cell biological processes, including anti-angiogenesis, inhibition of cell proliferation, suppression of immune responses, resistance to inflammatory responses, promotion of tissue integrity and quiescence, inhibit phagocytosis, and the synthesis of HA itself [15][16]. In contrast, the accumulation of LMW-HA (<250kDa) and HA oligosaccharides (<10 kDa) has been linked to the aggressiveness traits of cancer cells as it promotes angiogenesis, inflammatory responses, immune responses. It further induces the production of specific cytokines and enzymes necessary for the constitution of the tumour microenvironment (TME). Specifically, HA molecules around 200 kDa are known to trigger the release of inflammatory chemokines and impede the process of fibrinolysis [17,18]. Conversely, HA fragments smaller than 10 kDa enhance angiogenesis, facilitate cell migration, support the differentiation of endothelial cells, and also stimulate cytokine production in dendritic cells [13]. Additionally, HA fragments consisting of 4-6 oligo-saccharides (0.8-1.2 kDa) have been shown to activate the transcription of matrix metalloproteinases (MMPs) in both tumour cells and primary fibroblasts, thereby aiding in tumour progression [19]. In response to stress, tetrasaccharide chains of HA are capable of upregulating heat shock protein 72, thus preventing cell death [20]. In contrast to the promoting effect of HMW-HA on CD44 clustering, HA oligosaccharides disrupt this procedure [14].”

Authors: We have detailed the various effects of high molecular weight HA, low molecular weight HA, and HA oligosaccharides. However, as this information has been extensively reviewed elsewhere and is not the primary focus of this review, it is presented here without exhaustive detail.

Response to Suggestion 4: Since Section 3 (Uncontrolled HAIM in Solid Tumors) is too long, this section should be divided into at least two sections.

Authors: Thanks for your suggestion, section 3 is not that long when compared to other sections, and we can’t find an appropriate to split this section.

Response to Suggestion 5: Given the overall structure of this manuscript, it would be appropriate for section 8 to be before section 5.

Authors: Thank you for your revision, we think it is more appropriate to put section right after section 4.

Response to Suggestion 7: For ITIH, ACAN, VCAN, NCAN, and BCAN, the authors should explain each abbreviation when it first appears in the main text. On the other hand, for Neurocan, Aggrecan, and Versican, the abbreviations should be listed on page 8, line 236. Similarly, on page 11, lanes 397-399, the authors should explain ctDNA when it first appears in the main text.

Page 5 146-148: “Other molecules like TSG-6 and the extracellular matrix proteoglycan including HAPBs, HAPLN family and the lectican family (Neurocan (NCAN), Brevican (BCAN), Aggrecan (ACAN) and Versican (VCAN)) are secreted into the ECM [52-55].”

page 8, line 251: “similar modular structures such as CD44, NCAN, ACAN and VCAN [59,79]”

page 11 line 413-417: “DNA epigenetic analysis showed HAPLN3 methylated circulating tumour DNA(ctDNA) was widely found in de novo metastatic PCa (mPCa) and was markedly elevated in high-volume mPCa. Furthermore, the identification of methylated ctDNA was linked to a notably reduced period until the progression to metastatic castration resistant PCa [126].”

Authors: Thanks for your revision, these have been corrected.

Response to Suggestion 8: On page 5, line 135 and page 20, line 755, IAI and IaI should be IαI.

page 5, line 150: “Meanwhile, LYVE-1 is primarily expressed in lymphatic ECs[49], members of the IαI family are synthesised in the liver [56], and ACAN is synthesized primarily by chon-drocytes and other cartilage cells [57].”

page 20, line 821: “Apart from TSG-6, VCAN also binds to HCs of the IαI family and assists in the binding of SHAP to HA by attaching SHAP to GAG chains, however, such a promotive effect can be alleviated by MMP3-mediated degradation of VCAN [267,290].”

Authors: Thanks for your revision, these have been corrected.

Response to Suggestion 9: Please correct V3and V4 on page 10, line 380 and ECscan on page 16, line 570.

page 10, line 396: “Five human VCAN splice variants (V0, V1, V2, V3 and V4) have been characterized [174].”

page 17, line 624: “CD44 expressed by ECs can promote angiogenesis while an inhibition of CD44 warrants generation and stabilization of endothelial tubular network [241].”

Authors: Thanks for your revision, these have been corrected.

Response to Suggestion 10: For reference 1, a more appropriate citation is recommended.

Authors: The first reference on page 2 line 55 has been changed from ‘Two-faces’ of hyaluronan, a dynamic barometer of disease progression in tumor microenvironment to Dissecting the role of hyaluronan synthases in the tumor microenvironment. The new reference explores the crucial role of hyaluronan (HA) and its synthases (HAS1, HAS2, and HAS3) within the tumor microenvironment, focusing on how they influence cancer cell behavior and impact patient outcomes, and is more related to the topic of this review article. Although the previously cited references provide corresponding information, the changed citations are more relevant to the topic of this review and can give the reader more relevant information.

Reviewer 2 Report

Comments and Suggestions for Authors

Thank you very much for submitting your manuscript to CANCERS. Below are my review comments:

1. In the Conclusion section or end of the Discussion, the authors should add some description of HAIM's future perspective during metastasis and/or angiogenesis.

2. The authors should pay more attention to drawing figures; for example, in Fig. 7, cytokine, cell, macrophage should be changed to 'cytokines', 'cells', 'macrophages', respectively. Many tentative corrections are needed. 

Comments on the Quality of English Language

Minor editing is needed.

Author Response

Response to Suggestion 1: In the Conclusion section or end of the Discussion, the authors should add some description of HAIM's future perspective during metastasis and/or angiogenesis.

“HAIMs play diverse roles by interacting with HA and reshaping the TME during cancer progression. In general, their function is closely related with their subcellular distribution. Cancer cells, immune cells, mesenchymal cells, and various ECM components in the TME precisely regulate the synthesis and degradation of HA. The interaction between HAIMs and HA in turn affects the immune response, angiogenesis, and malignancy of tumour cells in the TME in a coherent and orchestrated fashion. The expression of some HAIMs is tissue-specific being lack of extensive and in-depth reports for their role in solid tumours.

Nanoparticles targeting RHAMM isoform B significantly reduced the tumour burden in a mouse model of pancreatic neuroendocrine tumour [309]. Short peptides with the HA-binding motif BX7B in RHAMM and CD44 have been shown to inhibit melanoma growth and induce apoptosis [310]. Other RHAMM mimic peptides have exhibited an inhibition of breast cancer cell invasion [311] and they can also induce prostate cancer cell apoptosis [312]. Studies also showed that a vaccination with RHAMM substantially diminished the tumour burden and enhanced the infiltration of RHAMM-specific cytotoxic T lymphocytes [313,314]. Subsequently, the effects of RHAMM-R3 peptide vaccine were tested in Phase I and Phase II clinical trials [315,316]. Administration of the RHAMM-R3 vaccine with chemotherapy or stem cell transplantation in patients with leukaemia can potentially delay relapse and maintain remission [316]. However, the multifunction of RHAMM and its specific expression in tissues have limited the application of these peptides. VCAN has been identified as a biomarker for tumour metastasis and chemotherapy resistance [303,317-319], assisting in the monitoring of off-target effects of platinum (Pt)-based chemotherapy and facilitating timely adjustments to therapeutic strategies [318]. Furthermore, VCAN is related to the efficacy of adjuvant chemotherapy, chemoradiotherapy, and immunotherapy in gastric cancer [320]. Targeting VCAN holds promise for enhancing the outcomes of current clinical treatments for gastric cancer.

However, the research on other molecules is limited due to their specific expression characteristics and low abundance in most tissues. Research related to LYVE-1 and cancer has primarily focused on its role in the lymphatic system and the immune responses it participates. However, recent studies increasingly indicate that LYVE-1+ macrophages also play a significant role within the TME [321-323]. Currently available anti-LYVE-1 antibodies allow for in vivo fluorescence imaging of lymphatic vessels and the disseminating tumour cells [324], and are used for positron emission tomography (PET) scanning [325].

NCAN can potentially serve as a target for glioma therapy and a prognostic marker [329,330]. Increased NCAN expression is associated with improved survival rates in Merkel cell carcinoma (MCC), though the specific mechanisms are yet to be elucidated [331]. Interestingly, NCAN-GFP fusion protein can specifically bind and illustrate the distribution patterns of HA in tissue sections and even live cells.

Taken together, this review underscores the reciprocal regulation between HAIMs and HA, providing a new perspective for a deeper understanding of molecular interactions in the TME. It is crucial for unveiling mechanisms of tumour development, discovering novel therapeutic targets, and designing more effective anticancer strategies.”

Authors: We added the current clinical applications and therapeutic potential of these molecules as treatment targets to the original review, and we have also discussed the shortcomings in current research.

Response to Suggestion 2: The authors should pay more attention to drawing figures; for example, in Fig. 7, cytokine, cell, macrophage should be changed to 'cytokines', 'cells', 'macrophages', respectively. Many tentative corrections are needed.

Authors: All singular and plural issues in picture 7 have been corrected. At the same time, the author also checked and ensured that there were no similar problems in other pictures.
